# Integrative Methylome and Transcriptome Characterization Identifies SERINC2 as a Tumor-Driven Gene for Papillary Thyroid Carcinoma

**DOI:** 10.3390/cancers15010243

**Published:** 2022-12-30

**Authors:** Tianxing Ying, Xumeng Wang, Yunjin Yao, Jimeng Yuan, Shitu Chen, Liping Wen, Zhijian Chen, Xiaofeng Wang, Chi Luo, Jinghao Sheng, Weibin Wang, Lisong Teng

**Affiliations:** 1Department of Surgical Oncology, The First Affiliated Hospital, Zhejiang University School of Medicine, Hangzhou 310000, China; 2Department of Thyroid Disease, The First Affiliated Hospital, Zhejiang University School of Medicine, Hangzhou 310000, China; 3Zhejiang Provincial Center for Disease Control and Prevention, Hangzhou 310000, China; 4Zhejiang Laboratory for Systems & Precision Medicine, Zhejiang University Medical Center, Hangzhou 310000, China; 5Affiliated Hangzhou First People’s Hospital and Institute of Environmental Medicine, Zhejiang University School of Medicine, Hangzhou 310000, China

**Keywords:** papillary thyroid carcinoma, DNA methylation, MethylationEPIC BeadChip (850K), RNA-Seq, SERINC2

## Abstract

**Simple Summary:**

Papillary thyroid carcinoma is still the most common endocrine tumor, most of which can be diagnosed by pathological results. However, there is still a small number of cases that are difficult to judge malignancy. In this study, we performed an integrative analysis of DNA methylation and RNA array from our cohort for potential papillary thyroid cancer-specific indicators. SERINC2, one of the differentially methylated and expressed genes, was first identified as a potential tumor-driven indicator in papillary thyroid carcinoma. We confirmed the influence of SERINC2 on proliferation and apoptosis in vitro after intervention or overexpression. Furthermore, we found tryptophan metabolism as a potential pathway targeted by SERINC2 through the investigation of data from the Cancer Dependency Map. In conclusion, our results demonstrate the whole-genome DNA methylation and gene expression profiles of papillary thyroid carcinoma, identify a new tumor-driven indicator, and provide a novel insight into the etiology of papillary thyroid cancer.

**Abstract:**

Most papillary thyroid carcinomas (PTCs) can be diagnosed preoperatively by routine evaluation, such as thyroid ultrasonography and fine-needle aspiration biopsy. Nevertheless, understanding how to differentiate indolent thyroid tumors from aggressive thyroid cancers remains a challenge, which may cause overtreatment. This study aimed to identify papillary thyroid cancer-specific indicators with whole-genome DNA methylation and gene expression profiles utilizing Infinium Methylation EPIC BeadChip (850k) and RNA arrays. In this paper, we report SERINC2 as a potential tumor-driven indicator in PTC. The up-regulated expression levels of SERINC2 were verified in PTC cell lines via qPCR. Then, cell counting kit 8 (CCK-8), wound healing, and flow cytometric assays were performed to confirm the influence of SERINC2 on proliferation and apoptosis in PTC cell lines after intervention or overexpression. Moreover, the investigation of data from the Cancer Dependency Map (DepMap) provided a potential pathway targeted by SERINC2. The activation of the tryptophan metabolic pathway may reduce the dependency of SERINC2 in thyroid cancers. In conclusion, our results demonstrate the whole-genome DNA methylation and gene expression profiles of papillary thyroid carcinoma, identify SERINC2 as a potential tumor-driven biomarker, and preliminarily verify its function in PTC.

## 1. Introduction

Papillary thyroid cancer (PTC) is the most common endocrine malignant tumor, with an increasing incidence worldwide over the past few decades [1,2]. In the past decades, some molecular genetic studies on PTC have been carried out, providing a deeper understanding of the incidence and progress of PTC. The activation of the MAPK pathway is widely known as a significant driver of PTC, including RET chromosomal rearrangement or point mutations of RAS or BRAF oncogenes, all of which trigger the activation of the mitogen-activated protein kinase (MAPK) cascade and the dysfunction of phosphatidylinositol-3 kinase (PI3K)/AKT signaling pathway, leading to change in cell growth and proliferation [3]. Therefore, mutations of MAPK pathway-related molecules may contribute to the incidence and development of thyroid cancer. Recently, with multi-platform analysis data, TCGA comprehensively displayed the genome map of PTC, extended the range of PTC driver alterations, and reduced the fraction of unknown oncogenic driver genes in PTC to 3.5% [4]. Recently, there is an increasing focus on the metabolism reprogramming in tumors [5,6], which is widely accepted as an essential role in tumor initiation and progression. Despite the research on the role of energy metabolism reprogramming in PTC [7,8], there are still unclear areas regarding the incidence mechanism of PTC, especially in amino acid metabolism. 

Papillary thyroid cancer has a good prognosis with a long-term survival rate of more than 90% [9]. Currently, the gold standard for distinguishing thyroid cancer from nodules is fine-needle aspiration (FNA) [10]; however, there is still a small number of cases that are difficult to judge with undetermined FNA cytologic results. Moreover, indolent clinical behavior with uncertain malignant potential is observed in some kinds of thyroid cancers. Thus, it is required to find an effective and convenient means to differentiate rapidly progressing from indolent lesions to avoid overtreatment.

DNA methylation, due to its role in regulating gene expression, has been described in different tumors including thyroid cancer in previous studies [11,12,13], and it has potency in the early diagnosis of tumors, molecular biomarkers, therapy, and prognostic prediction. Several studies have reported the potential use of methylation markers in distinguishing thyroid cancer from benign nodules in early clinical diagnosis [14,15]. However, previous studies have not investigated further the role of molecular markers in the progression of thyroid cancer.

In our study, the whole-genome DNA methylation and gene expression profiles between papillary thyroid carcinoma tissues and benign thyroid lesion tissues were compared and a number of differentially methylated and expressed genes were identified. Among these genes, serine incorporator 2 (SERINC2) was notable. SERINC2, belonging to the SERINC family, incorporates serine into membranes and plays an important role in facilitating the synthesis of phosphatidylserine and sphingolipids [16]. SERINC family is associated with a variety of diseases, including viral infections, cancers, neuro-related diseases, and other diseases [17]. SERINC2 has previously been reported to be up-regulated in a variety of cancers and has been considered a biomarker after multiple bioinformatics analyses for early diagnosis or prognosis prediction in lung adenocarcinoma [18,19], ovarian cancer [20], and low-grade glioma (LGG) [21]. Additionally, it has been proven that SERINC2 may affect cell proliferation and migration in lung adenocarcinoma cells through the PI3K/AKT pathway [22], and SERINC2 cis-activated by long non-coding enhancer RNA may regulate tumor lipid metabolism to affect leukemia progression in MLL rearrangement leukemia [23]. However, few studies about the role of SERINC2 in thyroid cancer have been conducted. 

Based on differences in methylation sites and RNA expression profiles between PTC and BTL samples, our study aimed to identify a potential new indicator with undiscovered functions in the progression of thyroid cancer, and preliminarily investigate its role in PTC.

## 2. Materials and Methods

### 2.1. Study Cohort and Design

Fresh frozen tissues were randomly chosen from 11 patients who underwent their first thyroidectomy in our medical center between 2016 to 2019 in this study (Appendix A). All tissue samples were retrieved from formalin-fixed paraffin-embedded blocks and reviewed by an experienced attending pathologist. Benign thyroid lesion (BTL) (n = 6) and papillary thyroid carcinoma (PTC) (n = 5) were sent for DNA methylation and transcriptome array The Institutional Ethics Committee approved the study. Informed consent was obtained from all patients.

### 2.2. DNA Differential Methylation Analysis and RNA Array

The DNA methylation profiles and RNA expression profiles of the samples in the cohort were performed utilizing Infinium Methylation EPIC BeadChip (850k) (Illumina, San Diego, CA, USA) and Affymetrix Clariom S array (Cat#00-00213, Affymetrix, Santa Clara, CA, USA), respectively, according to published protocols [15]. In brief, genomic DNA (500 ng) collected from BTL and PTC issues was isolated from the samples and treated with bisulfite using an EZ DNA Methylation Gold Kit (Zymo Research, Irvine, CA, USA). The methylation of DNA was assayed on the EPIC BeadChip (Illumina, San Diego, CA, USA) using the Illumina HD methylation assay kit from Shanghai Biotechnology Corporation. The original chip data were preprocessed by R software minfi package firstly, and then R software IMA package was used to screen the differences of methylation sites and methylation regions among the sample groups. The raw data were normalized, and the β-values were reported as a DNA methylation score ranging from 0 (completely unmethylated) to 1 (fully methylated). The formula for the beta value is listed as follows: $${\mathrm{beta}}_{i}=\frac{\max\left(y_{\left(i,\mathrm{methy}\right)},0\right)}{\max\left(y_{\left(i,\mathrm{methy}\right)},0\right)+\max\left(y_{\left(i,\mathrm{unmethy}\right)},0\right)+100}$$

Probes with absolute Δβ > 0.1 and detection *p*-values < 0.05 were considered as differentially methylated. The differentially methylated regions between sample groups were screened according to gene annotation classification and CpG island annotation classification. After the signal values of all probes in each region were taken as the arithmetic average, the pooled. *t*-test method was used to screen the differentially methylated regions in the sample groups (*p* value < 0.05 and |Beta.Difference| > 0.5). Then, the selected DMR-related genes were mapped to each term in the Gene Ontology or KEGG database, and the number of genes in each term was calculated. Then, hypergeometric test was applied to screen GO or KEGG entries significantly enriched in differential DMR-related genes compared with the whole genomic background, and the formula was calculated as follows:$$P=1-\overset{m-1}{\underset{i=0}{\sum }}\frac{\left(\begin{array}{c} M\\ i \end{array}\right)\left(\begin{array}{c} N-M\\ n-i \end{array}\right)}{\left(\begin{array}{c} N\\ n \end{array}\right)}$$

N is the number of genes with GO annotation among all genes. n is the number of differential DMR-related genes in N; M is the number of genes annotated with a particular GO or KEGG term among all genes; and m is the number of differential DMR-related genes annotated as a particular GO or KEGG term. A corrected *p*-value (FDR) ≤ 0.05 after Bonferroni correction was used as the threshold.

Total RNA was extracted and purified using RecoverAllTM Total Nucleic Acid Isolation (Cat#AM1975, Ambion, Austin, TX, USA) following the manufacturer’s instructions. Then, the total RNA was amplified, labeled, and purified to obtain biotin-labeled cDNA. Following array hybridization and washing using GeneChip® Hybridization, Wash and Stain Kit (Cat#900720, Affymetrix, Santa Clara, CA, USA) in a Hybridization Oven 645 (Cat#00-0331-220V, Affymetrix, Santa Clara, CA, USA) and a Fluidics Station 450 (Cat#00-0079, Affymetrix, Santa Clara, CA, USA), arrays were scanned by Affymetrix GeneChip® Scanner 3000 (Cat#00-00213, Affymetrix, Santa Clara, CA, USA) to obtain raw data. The raw data were normalized by Expression Console software of Affymetrix company and the fold changes and *p*-values were assessed to determine significance with the formula: foldchange = average (power (2, signal (FTC)))/average (power (2, signal (BTL))) and t-test separately. The parameter for Differentially expressed genes (DEGs) was set with fold change >2 or <0.5 and *p*-value < 0.05.

### 2.3. Cell Lines and Culture

The human papillary thyroid carcinoma cell lines B-CPAP and TPC-1 were purchased from DSMZ (Braunschweig, Germany), and human normal thyroid cell line Nthy-ori 3-1 was kindly provided by Dr. Tiannan Guo (School of Life Sciences, Westlake University). The cells were cultured in RPMI 1640 medium (Gibco, Rockville, MD, USA) supplemented with 10% fetal bovine serum (TIANHANG, Zhejiang, China) and 100 U/mL penicillin/streptomycin at 37 °C in a humidified incubator (PHCbi, Osaka, Japan) with 5% CO_2_.

### 2.4. Reagents and Transfection

Decitabine (AdooQ, A10292, stock concentration 50 mM), Ro 61-8048 (AdooQ, A12833), and Alizarin (AdooQ, A10054) were dissolved in DMSO and stored at −20 °C. Two short hairpin RNAs (shRNAs) targeting SERINC2 were designed and inserted into the pLKO.1 puro vector. The coding sequence of SERINC2 was cloned into the pcDNA3.1 vector. The following sequences are shown in Appendix A. Cells were transfected using Lipofectamine 2000® (Invitrogen; Thermo Fisher Scientific, Inc., Waltham, MA, USA), according to the manufacturer’s protocols. 

### 2.5. Cell Proliferation Assay

In total, 3000 cells per well were seeded into 96-well-plates with 100 μL full medium and incubated for 6 hours, 1 day, 2 days, and 3 days at 37 °C. Subsequently, 10 μL CCK-8 reagent (Dojindo, Kumamoto, Japan) was added to each well and indicated for 2–3 h at 37 °C. The absorbance was detected by spectrophotometer (Thermo Fisher, Waltham, MA, USA) at a wavelength of 450 nm. The data were analyzed using GraphPad Prism 8.

### 2.6. Cell Apoptosis Analysis

Apoptosis was performed by flow cytometry measurement. Briefly, after being transfected with shRNA or negative control vector for 24 h, cells were harvested and gently washed with cold PBS and stained with Annexin V-APC and PI staining buffer (YEASEN, Shanghai, China) for 15 min incubation in the dark, according to the protocols. At last, cell apoptosis was detected by flow cytometry (ACEA Biosciences, San Diego, CA, USA) and analyzed by the Novo Express software.

### 2.7. RNA Extraction and Real-Time Quantitative RT-PCR (qRT-PCR) Assay

Total RNA was extracted using the TRIZOL reagent (TAKARA, Kusatsu, Japan). First-strand cDNA was generated from 1-2 µg of total RNA using a commercial SuperScript II First Strand Synthesis System (TAKARA, Kusatsu, Japan). Real-time PCR was performed using SYBR Green PCR Master Mix (TAKARA, Kusatsu, Japan) in a Step One Plus RT-PCR system (Roche, Basel, Switzerland). The amplified transcript level of each specific gene was normalized to β-actin using the 2^−ΔΔCt^ method. The sequences of primers used in this study are listed in the Appendix A.

### 2.8. Western Blotting

For protein analysis, cells were seeded into flat-bottomed 6-well plates for the indicated time (2 × 10^6^ cells/well). Cells were washed with PBS and lysed with RIPA buffer supplemented with 1% phenylmethylsulfonyl fluoride (PMSF), 1% protease inhibitor cocktail, and 1% sodium orthovanadate (Biosharp, Beijing, China), followed by the determination of total protein concentration using the BCA kit (Beyotime, Shanghai, China) and samples were subjected to Western blotting on a denaturating 10% acrylamide gel. Membranes were blocked with 5% BSA in PBS and blotted with primary antibodies as follows: Anti-SERINC2 (Proteintech, 20266-1-AP), and beta-actin (Proteintech, HRP-60008) used as loading controls. After washing the membrane with TBST, HRP-labelled goat anti-rabbit IgG (H+L) in 5% BSA in TBST was added to the membrane and incubated for 1 h at room temperature. After washing with TBST, signals were visualized using the enhanced chemiluminescence detection system by Amersham Imager 600 (GE Healthcare Life Sciences, Pittsburgh, PA, USA).

### 2.9. The Cancer Dependency Map Database

The Cancer Dependency Map (https://depmap.org/portal/, accessed on 28 October 2022) is a website that can predict cancer dependencies with a specific algorithm based on genome-scale CRISPR-Cas9 essentiality screens of hundreds of cell lines models [24]. CERES and Chronos are different algorithms to estimate gene dependency for identifying the genetic vulnerabilities of cancer based on CRISPR-Cas9 screens [25,26].

### 2.10. Statistical Analysis

Quantitative data are presented as mean ± SD of at least three experiments. Data were analyzed using GraphPad Prism 8. Differences between the two groups were assessed with the Student’s *t*-test. One-way ANOVA was used for multiple comparisons to test for significant differences. *p* < 0.05 was considered to indicate a statistically significant difference. The analysis of Gene Ontology (GO) and KEGG (Kyoto Encyclopedia of Genes and Genomes) enrichment was conducted using the R package clusterProfiler.

## 3. Results

### 3.1. Hypomethylated and Up-Regulated Genes Predominate in Papillary Thyroid Carcinoma

DNA methylation and RNA transcriptome array were performed using a methylation chip (850K) and gene expression profile chip, respectively with the select tissue samples from five papillary thyroid carcinoma (PTC) and six benign thyroid lesions (BTL). The flow chart of the exploration of methylation-driven genes is shown in Figure 1A. The differentially methylated regions among sample groups were screened according to gene annotation classification and CpG island annotation classification. The gene annotation classification included TSS1500, TSS200, 5’ UTR, 1stExon, genebody, and 3’ UTR. The CpG Island annotation classification contains five regions: N shelf, N shore, CpG Island, S shore, and S Shelf. The hypomethylation regions were mainly located in the N shelf and S shelf, while hypermethylation regions were mainly located in CpG island and S shore. From the point of view of gene annotation distribution, hypomethylated regions were mainly located in TSS 1500 and gene body, while the distribution of hypermethylated regions was mainly located in 1stExon (Figure 1B). Compared with the control sample, 483 genes were significantly lowly methylated and 83 genes were high methylated in PTC samples, suggesting that most of the genes show a low methylation state(|Δβ| > 0.1, *p* < 0.05) (Figure 1C,D, Appendix A). We performed KEGG pathway analysis of the differential methylated genes, and the top 10 significant pathways are displayed. *Peroxisome* and *Phospholipase D signaling pathway* were enriched for the hypermethylated genes, while *cytokine-cytokine receptor interaction*, *cAMP signaling pathway* and *Cell adhesion molecules* were enriched for the hypomethylated genes (Figure 1E).

Moreover, the transcriptome results of the same samples show that, compared with the control sample, 167 genes were up-regulated and 290 genes were down-regulated in PTC samples (log|FC| > 1, *p* < 0.05) (Figure 2A). We conducted a GO functional enrichment analysis and a KEGG pathway enrichment analysis for the further understanding of potential functions. The results showed that *Cell adhesion molecules*, *fluid shear stress,* and *atherosclerosis* were enriched for the up-regulated genes, while *thyroid hormone synthesis* and *adrenergic signaling in cardiomyocytes* were enriched for the down-regulated genes (Figure 2B). The GO analysis results suggested that down-regulated genes are involved in *ATP binding*, *plasma membrane* and *cytoplasm* (Figure 2C), while the *extracellular region* and *extracellular exosome* are involved with the up-regulated genes (Figure 2D).

Integrating the abnormal methylation expression and gene expression data, the expression of 91 genes may be driven by methylation, among which 65 genes were hypomethylated and highly expressed, and 26 genes were hypermethylated and lowly expressed (Figure 3A). Most genes showed high expression and hypomethylation, which was consistent with the trend of results verified in TCGA (Appendix A). According to the difference in methylation and expression, we derived some genes with high methylation, such as DIO1, RAPGAP, ERRFI1, and BMP8A. After being treated with different doses (1, 2.5, and 5 μM) of 5-aza-2′-deoxycytidine, a demethylating agent, for 72 h, the expression of the chosen genes of TPC-1 was up-regulated with increasing concentrations (Appendix A), supporting the relationship between methylation and the expression of selected genes. The KEGG results show that the hypermethylated and down-regulated genes are involved in *thyroid hormone synthesis* and *cardiac muscle contraction*, while the hypomethylated and up-regulated genes are related to *cell adhesion molecules (CAMs)* (Figure 3B,C). The GO analysis results suggested that hypermethylated and down-regulated genes were involved in *calcium ion binding* and *Golgi membrane* (Figure 3D), while the *plasma membrane* and *extracellular region* were involved in the hypomethylated and up-regulated genes (Figure 3E). 

### 3.2. SERINC2 Is Up-Regulated in Thyroid Cancer

According to log2|FC| being greater than 1 and the delta-beta value being greater than 0.1 or lower than -0.1, we selected CYP1B1, DIO1, FABP3 and SERINC2 as potential tumor-promoting genes through metabolic reprogramming, as the intersection of metabolism pathway related genes (https://pathcards.genecards.org/Card/metabolism, accessed on 28 October 2022) and the genes with significant differential methylation and expression (Figure 4A, Appendix A). To focus on the most relevant gene to thyroid cancers, using Chronos dependence scores of cell lines from different organs by CRISPR-Cas9 screening obtained from DepMap, SERINC2 was found to be more dependent in thyroid cancer cell lines than in other organ cell lines (Figure 4B), suggesting that SERINC2 may play a more important role in thyroid cancer. Compared with the benign control, SERINC2 is hypomethylated and highly expressed in PTC. Therefore, we verified that SERINC2 was highly expressed in thyroid cancer cell lines (Figure 4C). Expression data from the TCGA dataset further showed that SERINC2 was up-regulated in thyroid cancer tissues compared with normal tissues (Figure 4D). Next, we performed the immunohistochemical staining of SERINC2 for verifying the expression using five paraffin-embedded PTC samples (Figure 4E). ROC analysis with TCGA plus GTEx datasets indicated that the diagnostic value of SERINC2 was high (AUC > 90%) in thyroid cancers (Figure 4F. Survival analysis using GEPIA2 suggested that the high expression of SERINC2 is associated with worse DFS (Figure 4G), but the expression of SERINC2 is not associated with overall survival (Appendix A). 

### 3.3. Regulation of SERINC2 Influences Papillary Thyroid Cancer Progression

To explore the significance of SERINC2 in PTC cells, SERINC2 was overexpressed by transfection with a pcDNA3.1 plasmid (Figure 5A). The proliferation of papillary thyroid cancer TPC-1 cell lines was promoted after SERINC2 overexpression (*p* < 0.05, Figure 5B). CCK-8 assay was used to examine cell viability. To identify the effect of SERINC2 silencing on the proliferation of PTC cells, SERINC2 expression in TPC-1 was significantly suppressed by shRNA (*p* < 0.05, Figure 5C). SERINC2 knockdown significantly suppressed the proliferative activity and viability of TPC-1, assessed using CCK-8 assay and cell count (Figure 5D). Furthermore, apoptosis was induced in TPC-1 when SERINC2 expression was reduced (Figure 5E). These results indicated that SERINC2 would be relevant to PTC cell proliferation.

### 3.4. Activation of Tryptophan Metabolic Pathways Reduces the Dependence on SERINC2 in Thyroid Cancer

To find the related mechanism of SERINC2 in thyroid cancer, we filtered the related genes that may substitute SERINC2 functionally based on the data from Depmap. CERES and Chronos are similar but not the same algorithms to estimate gene-dependency levels from CRISPR-Cas9 essentiality screens. Those genes with higher expression and lower dependency on SERINC2, which means gene expression is positively correlated with the SERINC2 gene effect, may partly substitute SERINC2 functionally. Functional enrichment analysis was conducted using the positively correlated genes selected by CERES and Chronos (Figure 6A). The intersection of the CERES and Chronos datasets contained 505 genes (Figure 6B). The function enrichment analysis revealed that multiple metabolic pathways may be affected when SERINC2 was knocked out, especially the tryptophan metabolism (Figure 6C). The expressions of IDO1, TDO2, KMO, and CYP1B1 in the tryptophan metabolism were positively correlated with the SERINC2 gene effect (Figure 6D,E). These genes may help cells to survive when SERINC2 is knocked out. To confirm this, we treated TPC-1 with inhibitors of IDO/TDO, KMO, and CYP1B1 while SERINC2 was knocked down with shRNA, and found that the inhibition of both SERINC2 and tryptophan metabolism enhanced cell apoptosis (Figure 6F).

## 4. Discussion

PTC is the most common type of thyroid cancer. Despite the known risk factors for PTC including radiation exposure, iodine uptake, obesity, and the deep research on the molecular etiology of BRAF, RAS, and RET mutations, the mechanism of the incidence and development of PTC is still unknown. Due to the low distant metastasis rate and low mortality rate, most PTC patients have an excellent prognosis, but there is still a small part with stronger invasiveness and poorer prognosis. How to identify further the indolent ones and avoid excessive diagnosis and treatment, is currently a critical challenge. In the present study, we demonstrated the DNA methylation and gene expression profiles in papillary thyroid carcinoma samples and benign thyroid lesion samples, providing more evidence on the mechanism of the incidence and development of PTC. Additionally, the function of SERINC2 in PTC was preliminarily explored in this study. SERINC2 was found to have a potential relationship with the regulation of tryptophan metabolism in PTC, suggesting that tryptophan metabolism is likely involved in the mechanism of PTC.

DNA methylation plays an essential role in the regulation of gene expression and silencing, as one of the most abundant epigenetic modifications, contributing to the pathogenesis of cancer and other diseases, including thyroid carcinoma [27,28,29]. Indeed, for its stability and non-invasiveness, DNA methylation has the potential to be a biomarker for diagnosis and prognosis. Previously, a few studies have used DNA methylation profiling and transcriptome array to select potential genes or methylation sites in thyroid cancer as diagnostic or prognostic biomarkers [15,30,31]. In our study, we found that the DNA methylation level was generally lower in PTC than in BTL. Additionally, by combining DNA methylation with gene expression data, we further found a relationship between the expression and methylation level in some genes, such as RAP1GAP [32] and BMP8A [33], which proved to play a role in the progression of thyroid cancer. 

Furthermore, we investigated SERINC2, hypomethylated and highly expressed in PTC, for the function in cell lines. SERINC2 belongs to SERINC family, and participates in sphingolipid biosynthesis. In several previous studies, SERINC2 was identified as a biomarker in different tumors [18,19,20,21], for its different expression in tumors and non-tumor tissues. Our study further analyzed SERINC2 expression in PTC, and demonstrated that its aberrant expression in PTC is driven by epigenetic changes. Furthermore, we evaluated the potential diagnosis and prognosis values of SERINC2 by analyzing a variety of databases, and verified its correlation with clinical features. Distinguishing PTC from normal tissues according to the expression of SERINC2 showed a high diagnostic value, with a sensitivity of 93.2% and a specificity of 98.8%. Additionally, the high expression of SERINC2 was found to be associated with worse disease-free survive, suggesting that expression of SERINC2 may be used for prognosis prediction. However, the expression level of SERINC2 did not show significant difference at different pathologic stages in PTC, probably due to the small size of advanced papillary thyroid cancer samples. With a good predictive ability in the diagnosis and prognosis of PTC, SERINC2 may play an important role in the growth and progression of PTC, and has the potential to serve as a diagnostic and prognostic biomarker.

Researchers have been exploring the ways in which SERINC2 influences the progress of cancers. Zeng et al. reported that SERINC2-knockdown may inhibit the proliferation and migration of lung adenocarcinoma cells via regulating the PI3K/AKT pathway [22]. Ke Fang et al. demonstrated that SERINC2, *cis*-activated by lnc-eRNA, influences the biosynthesis of sphingolipids to promote leukemia progression^23^. Furthermore, our study linked the high gene expression of SERINC2 in PTC with hypomethylation in the promoter region. Our study also revealed that the high expression of SERINC2 promotes PTC cell progress while SERINC2 silencing may induce cell apoptosis and inhibit PTC proliferation. It was also reported that SERINC2 could inhibit apoptosis by activation of Akt in acute lung injury [34]. However, it is unclear how SERINC2 can activate Akt, which needs further experimental proof. Moreover, we used Depmap data to explore the potential pathway related to apoptosis induced by SERINC2-knockdown. Tryptophan metabolism through the kynurenine pathway has been proven to have an important role in different cancers [35,36]. It has been found that the tryptophan metabolism pathway may be related to cell survival and apoptosis, as the recent report showed that activation of the tryptophan–kynurenine–AHR pathway promotes leiomyoma cell survival and decreases apoptosis [37]. The consumption of tryptophan and accumulation of metabolite by microorganisms [38] can interact with intestinal epithelial cells, affect the host intestinal barrier, and act on a variety of tissues and organs through the circulatory system. These metabolites, on the other hand, can promote tumor cell growth and limit immune cell activation through AhR and PXR receptors. Nevertheless, whether it functions in thyroid carcinoma remains largely unknown, despite the tryptophan metabolite differences between PTC and the healthy control group observed in metabolomics [39,40]. Therefore, SERINC2-knockdown has been found to induce apoptosis in PTC cells and is likely associated with tryptophan metabolism. Additionally, simultaneous inhibition of SERINC2 and tryptophan metabolism may result in a synergistic effect in apoptosis. Further work is needed to confirm the relationship between SERINC2 and tryptophan metabolism. Additionally, in vivo studies are required to prove the role of SERINC2 in PTC.

## 5. Conclusions

In summary, our work shows a preliminary integrated analysis of the methylome and transcriptome in PTC, and a novel perspective for SERINC2 as a potential biomarker to tell indolent thyroid tumors from more aggressive thyroid cancer.

## Figures and Tables

**Figure 1 cancers-15-00243-f001:**
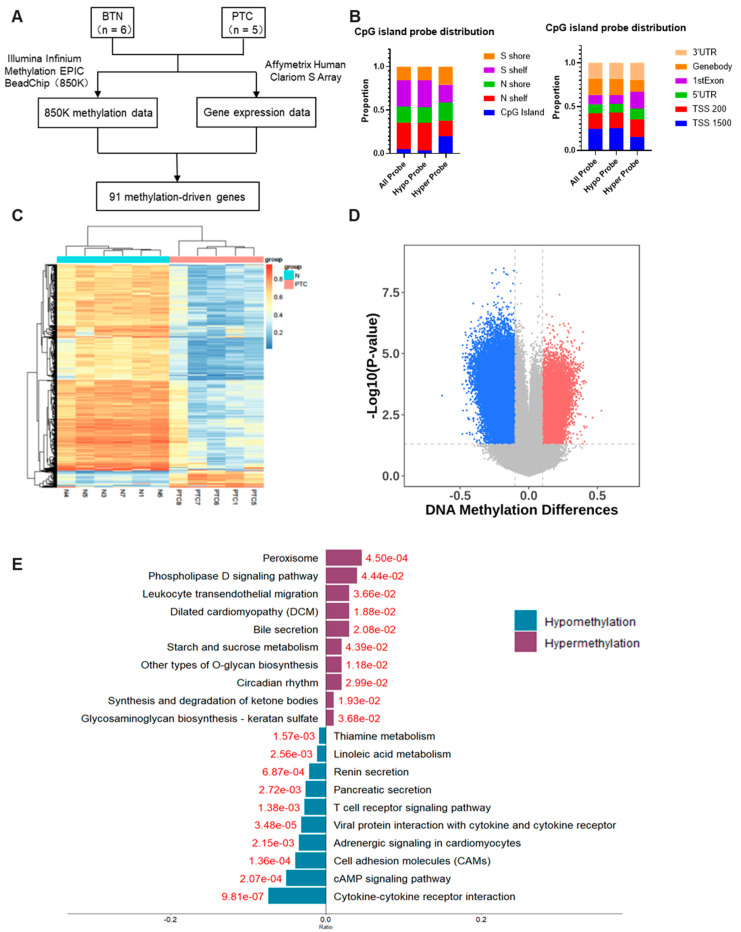
The whole-genome DNA methylation profile of papillary thyroid carcinoma. (**A**) The schematic flow chart of selection of methylation-driven genes. (**B**) Differential methylation regions distribution according to CpG island annotation, and according to gene annotation. (**C**,**D**) DNA methylation differences between thyroid nodules and PTC are shown by a heatmap (**C**) and a volcano plot (**D**). (**E**) The top 10 KEGG enrichment analysis results of genes with significant methylation differences.

**Figure 2 cancers-15-00243-f002:**
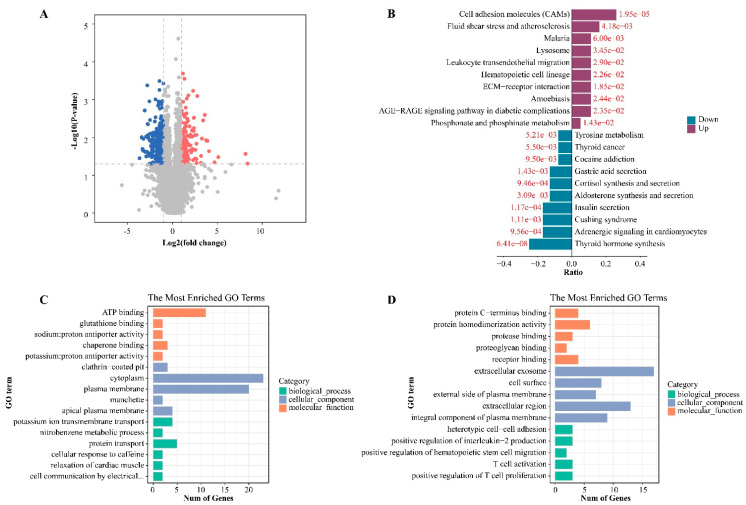
The whole-genome transcriptome profile of papillary thyroid carcinoma. (**A**) Volcano plot showing transcriptome difference between thyroid nodules and PTC. (**B**) The top 10 KEGG enrichment analysis results of genes with significant expression differences. (**C**) The top 5 GO enrichment analysis results of significantly down-regulated genes. (**D**) The top 5 GO enrichment analysis results of significantly up-regulated genes.

**Figure 3 cancers-15-00243-f003:**
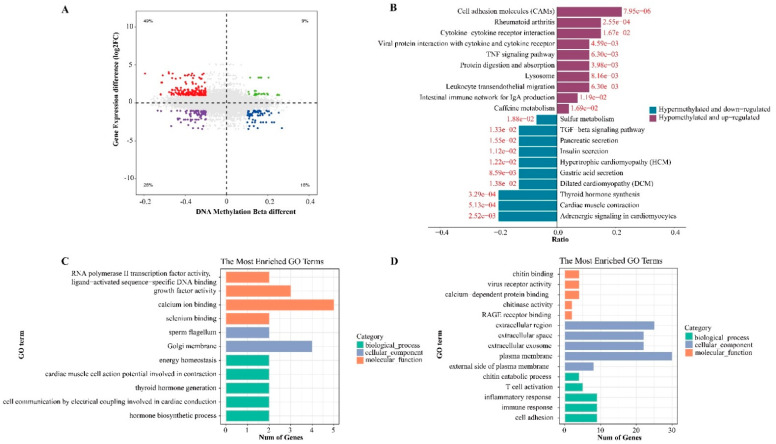
Selection of methylation-driven genes in thyroid papillary carcinoma. (**A**) Volcano plot showing the integrative analysis of the methylation profile and transcriptome profile of PTC. Red dots represent hypomethylated and highly expressed genes, and blue dots represent hypermethylated and lowly expressed genes. (**B**,**C**) The top 10 KEGG enrichment analysis results of hypermethylated and down-regulated genes and hypomethylated and up-regulated genes. (**D**) The top 5 GO enrichment analysis results of hypermethylated and down-regulated genes and hypomethylated and up-regulated genes.

**Figure 4 cancers-15-00243-f004:**
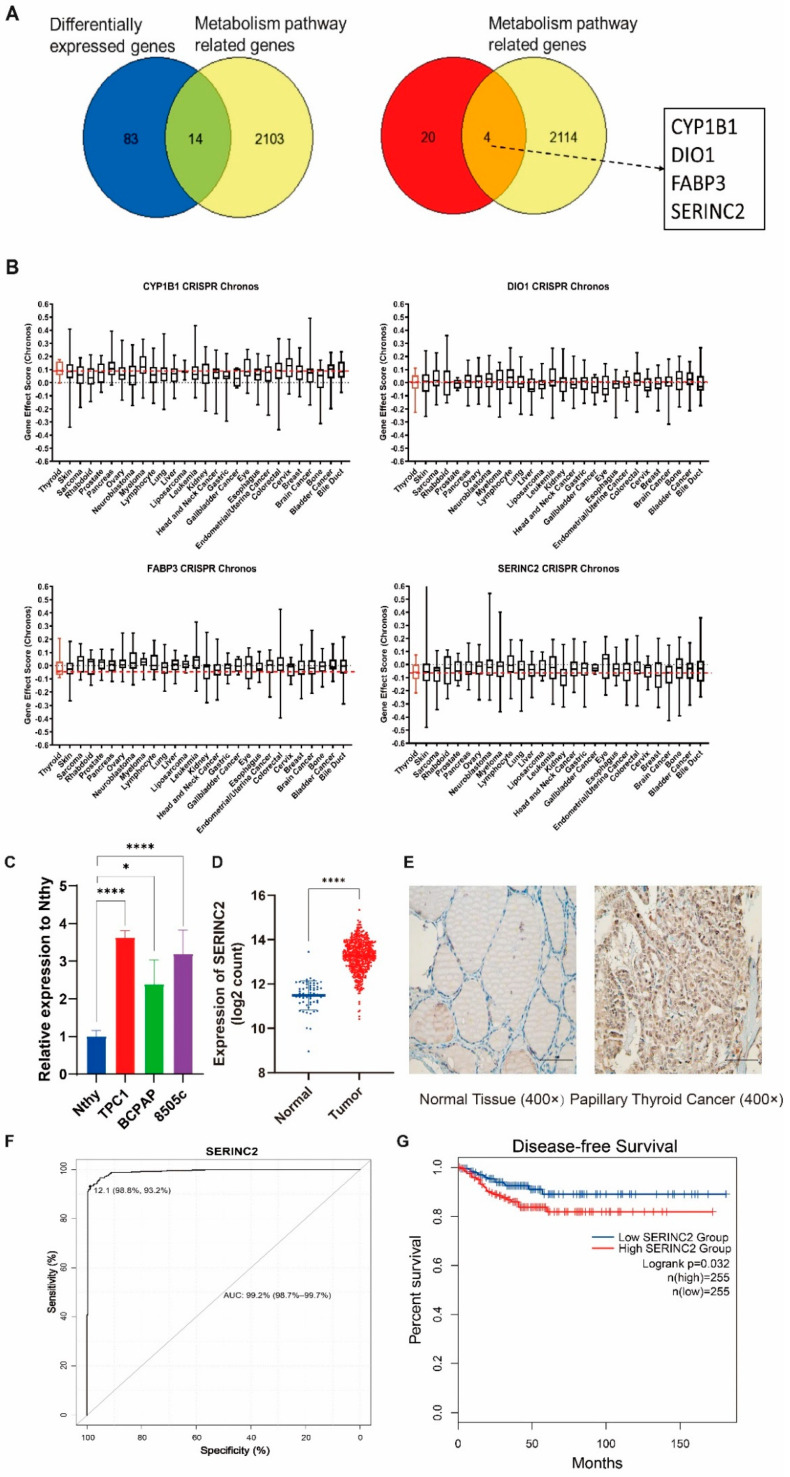
SERINC2 is up-regulated in thyroid cancer. (**A**) The intersection of metabolism pathway related genes and all the differentially expressed genes on the left, and the intersection of metabolism pathway related genes and the selected genes with significant differential methylation and expression from the methylation profile and transcriptome profile of PTC on the right. (**B**) Chronos dependence scores of CYP1B1, DIO1, FABP3 and SERINC2 by CRISPR-Cas9 screening from Depmap in cell lines derived from different organs. (**C**) Expression of SERINC2 in different thyroid cancer cell lines. (**D**) Expression analysis using GEPIA2 shows SERINC2 is up-regulated in thyroid cancer tissue. (**E**) Representative immunohistochemical pictures of PTC and adjacent tissues (400×). (**F**) ROC analysis of SERINC2 expression in thyroid cancer from TCGA and GTEx databases. (**G**) Survival analysis shows high expression of SERINC2 is associated with worse DFS. Quantitative results are shown in bar graph as mean ± SEM. * *p* < 0.05 and **** *p* < 0.001.

**Figure 5 cancers-15-00243-f005:**
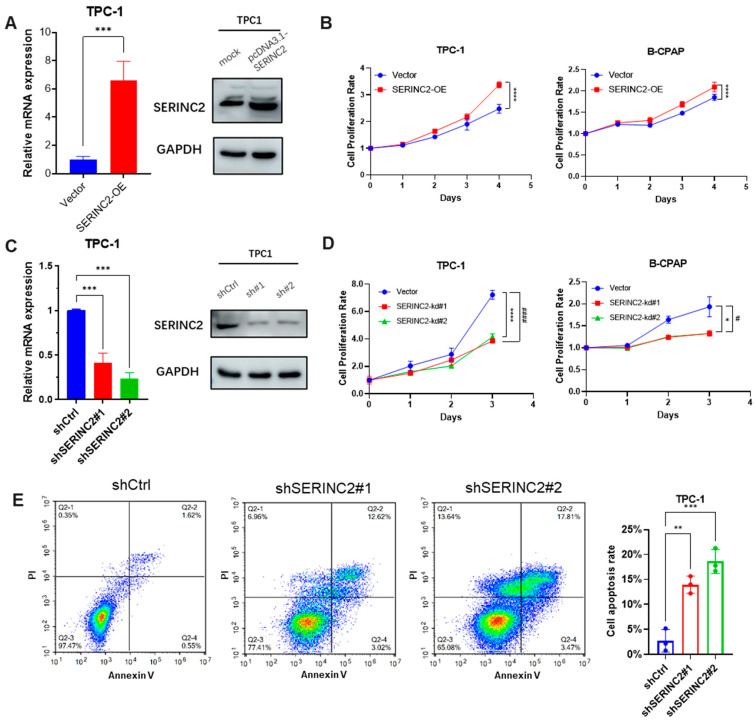
Regulation of SERINC2 influences papillary thyroid cancer progression. (**A**) Relative mRNA expression levels of SERINC2 in TPC-1 were significantly up-regulated by transfecting pcDNA3.1 plasmids for 48 h. (**B**) TPC-1 cells were transfected with pcDNA3.1 plasmids, followed by measurements of cell growth by CCK-8. (**C**) mRNA levels of SERINC2 were knocked down with shRNA in TPC-1 cells. (**D**) Knockdown of SERINC2 with shRNA plasmids disrupted the cell proliferation in TPC-1 cells. (**E**) Apoptotic levels in TPC-1 cells transfected with shRNA were determined by a flow cytometer. Results are presented as the mean ± SEM. *, # *p* < 0.05, ** *p* < 0.01, *** *p* < 0.005 and ****, #### *p* < 0.001.

**Figure 6 cancers-15-00243-f006:**
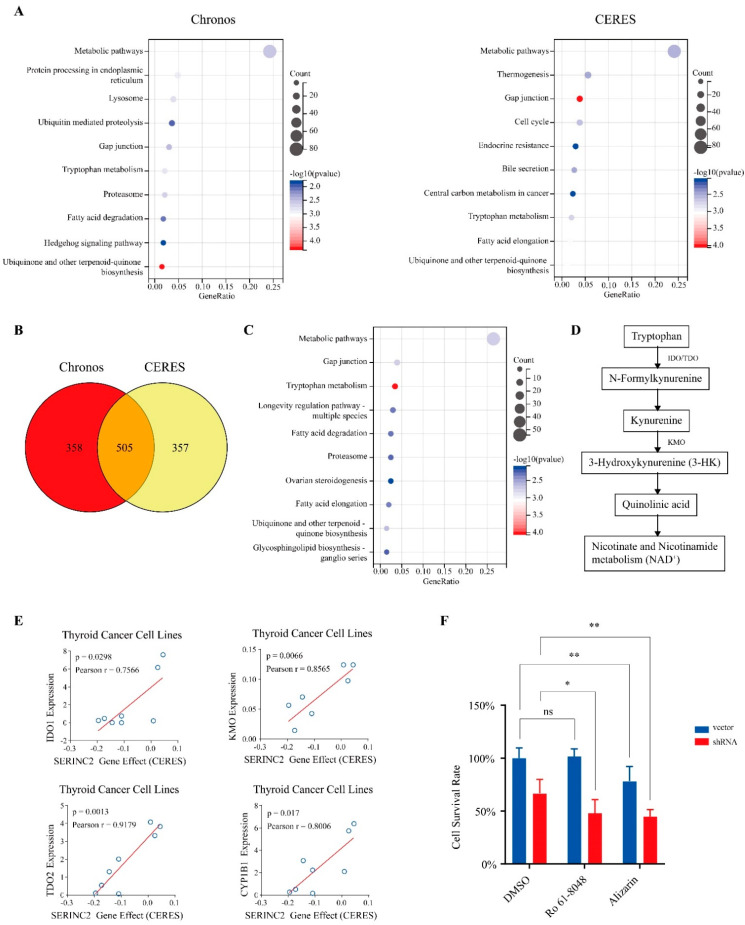
Activation of tryptophan metabolic pathways reduces the dependence on SERINC2 in thyroid cancer. (**A**) KEGG enrichment analysis results of genes selected by CERES and Chronos. (**B**,**C**) The intersection and KEGG enrichment analysis result of gene sets selected from CERES and Chronos. (**D**) Brief schematic diagram of the tryptophan-kynurenine metabolic pathway from KEGG. (**E**) The fitting line for the gene expression and SERINC2 gene effect of IDO1, TDO2, KMO, and CYP1B1. (**F**) TPC-1 cells were treated with 20nM of the KMO inhibitor Ro 61-8048 and 5µM of CYP1B1 inhibitor Alizarin after being transfected with shRNA for 24 h, followed by viability measurement with CCK8. Quantitative results are shown in bar graph as mean ± SEM. * *p* < 0.05, ** *p* < 0.01.

## Data Availability

The data presented in this study are available in the paper and/or the Appendix A. Additional data related to this paper are available on request from the corresponding author.

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
