# Peer review of "Integrative Methylome and Transcriptome Characterization Identifies SERINC2 as a Tumor-Driven Gene for Papillary Thyroid Carcinoma"

_cancers, 2022, doi:10.3390/cancers15010243_

Round 1

Reviewer 1 Report

There are number of specific points to address:

1.       English editing is required

2.       Introduction is not focused. It is not clear why SERINC2 has been chosen for analysis.

3.       There are sentences that are difficult to understand. An example “However, there are still unclear areas regarding the incidence mechanism of PTC, especially in metabolism, while most related studies focused on energy metabolism reprogramming”

4.       This study was performed using samples from 11 patients (6 with benign and 5 with malignant thyroid tumors). The number of examined samples in not sufficient to drive a valid conclusion.

5.       Please provide more detailed information on the method “DNA Differential Methylation Analysis and RNA Sequencing”

6.       Please include in Method section information of Western blot analysis.

7.       Please provide more clarification on the following sentence: “The hypomethylation sites were mainly located in the N shelf and S shelf, while hypermethylation sites were mainly located in CpG island and S shore”.

8.       Please provide information on the effects of SERINC2 overexpression and SERINC2-knockdown in BCPAP cells.

9.       Please provide more data demonstrating that inhibition of both SERINC2 and tryptophan metabolism enhanced cell apoptosis.

10.   Please discuss in more detail the potential role of tryptophan metabolism in thyroid cancer.

Author Response

Thank you for your valuable comments. A point-by-point response is listed below. Please see the attachment.

Reviewer 2 Report

    In their manuscript entitled “Integrative methylome and transcriptome characterization identifies SERINC2 as a tumor-driven gene for papillary thyroid carcinoma (PTC), the authors Ying et al. perform a transcriptomic and whole-genome methylome analysis on paraffin embedded tissue from cohorts of patients who underwent undergoing thyroidectomy for potential papillary thyroid carcinoma.  The tumor expression and methylation patterns were compared to benign thyroid lesions and a bioinformatics analysis was performed to determine differentially methylated and expression genes, particularly focusing on genes involved in metabolic pathways.  Their results indicated that SERINC2, previously identified as misregulated in a host of pathologies including various cancers was upregulated in PTC relative to benign lesions. Verification of upregulation and SERINC2’s role in cancer cell proliferation was verified using shRNA studies in a panel of PTC cell lines, which overexpress SERINC2.  There appears to be a gene dosing effect of SERINC2 on cell proliferation.

In general, the paper is readable and well written.  However there are a few major concerns with this study which need to be addressed before it could be accepted for publication in Cancers.

A. Materials and Methods -  The authors need to describe, in more detail, how the tissues were analyzed, including how they determined that 11 patient samples were enough power for this study, stratify the samples according to histo-clinical grade and stage, give a table of patient characteristics, and finally describe how the tissues were analyzed (was laser capture performed to select areas of the sections?)  Also it would be a very good idea to either show some immunohistochemistry of your tissue or even an IHC with SERINC2 would be great as well.  Also was this a cancer versus benign lesion or did you analyze some normal adjacent tissue?

For the methylation analysis please provide the sensitivity and specificity for the bisulfite assay.  What was the false positive rate?

For expression analysis there is a major concern of obtaining quality RNA from FFPE tissues and there should be an analysis of the quality of RNA used in microarray.

In section 2.5 for cell proliferation pleas in line 124 state what the units for incubation is (hours, days?)

In statistics section you need to supply some statistical analysis or power calculations on how you performed the microarray and how did you determine how many replicates you needed.

B. Results -

First could there be a Venn diagram with all the differentially expressed genes and then you can show just the genes involved in metabolic pathways.  I am curious why you focused on these and not also those involved in adhesion as well (as you showed in Figure 3B).

In figure 5A, the western blot does not seem to correlate with the densitometry.  Are the lanes reversed?.  In Figure 5B there is not much of an effect, even at 4 days.  Perhaps another method like cell counting would be a better indication of proliferation effect and a calculation of doubling time for each condition. In addition maybe a clonogenic assay would be more suitable.

In figure 5 C and D are you suggesting there is a gene dosing effect of knockdown of SERINC2 as the one knockdown is not complete.  It may explain the results in D.  The results in E look like late stage apoptosis with more necrotic cells (high PI staining).  Please show also compensation controls.

In Table 2 please explain why you used the cutoff at log[FC]>1.

Author Response

(The authors gave the same response as above.)

Reviewer 3 Report

This manuscript reports an interesting topic, integrating papillary thyroid carcinoma's methylation and expression profile (PTC) to identify the candidate gene related to this cancer type. However, some parts are still unclear in the current manuscript, especially the mechanism of SERINC2  in PTC. The Experiment design for the study of the mechanism of SERINC2 was not well designed.  DNA methylation was mentioned but did not develop in the manuscript's later part. And the figures in the manuscript must be re-format to the same font and size.

Author Response

(The authors gave the same response as above.)

Round 2

Reviewer 1 Report

English editing is required. Certain sentences are just impossible to understand.   

Author Response

Thank you for underlining this deficiency. We have worked on both language and readability and have submitted our manuscript to MDPI for the paid English editing. Please see the attachment.

Reviewer 2 Report

Thank you for your answers and providing the corresponding changes to the manuscript.  I do believe a longer term assay such as colony formation in soft agar or even just a clonogenic assay on plastic or just a FACS analysis of cell cycle would strentgthen your point of SERINC2 affecting cell proliferation.  There also may be some interesting insights into mechanisms how SERINC2 promotes cell proliferation or promotes survival, as this reviewer is thinking.  It also would be good to get that sensitivity and specificity data from the assay.  Commentors would need to see that given the lower sample size you have.

Author Response

Thank you for your constructive comments on my manuscript. Please see the attachment.

Reviewer 3 Report

Please see the attached file for specific comments

Author Response

(The authors gave the same response as above.)

Round 3

Reviewer 1 Report

Authors have addressed questions.